# Single versus dual-rate learning when exposed to Coriolis forces during reaching movements

**Judith L. Rudolph**[1], **Janny C. Stapel**[2], **Luc P. J. Selen**[1], **W. Pieter Medendorp**[1] *

**1** Donders Centre for Cognition, Donders Institute for Brain, Cognition and Behaviour, Radboud University, Nijmegen, The Netherlands, **2** Department of Psychology, Uppsala University, Uppsala, Sweden

* p.medendorp@donders.ru.nl

**Data Availability Statement:** Data are available from the Donders Institute for Brain, Cognition and Behavior at: https://doi.org/10.34973/86sa-gx85.

## Abstract

When we reach for an object during a passive whole body rotation, a tangential Coriolis force is generated on the arm. Yet, within a few trials, the brain adapts to this force so it does not disrupt the reach. Is this adaptation governed by a single-rate or dual-rate learning process? Here, guided by state-space modeling, we studied human reach adaptation in a fully-enclosed rotating room. After 90 pre-rotation reaches (baseline), participants were trained to make 240 to-and-fro reaches while the room rotated at 10 rpm (block A), then performed 6 reaches under opposite room rotation (block B), and subsequently made 100 post-rotation reaches (washout). A control group performed the same paradigm, but without the reaches during rotation block B. Single-rate and dual-rate models can be best dissociated if there would be full un-learning of compensation A during block B, but minimal learning of B. From the perspective of a dual-rate model, the un-learning observed in block B would mainly be caused by the faster state, such that the washout reaches would show retention effects of the slower state, called spontaneous recovery. Alternatively, following a single-rate model, the same state would govern the learning in block A and un-learning in block B, such that the washout reaches mimic the baseline reaches. Our results do not provide clear signs of spontaneous recovery in the washout reaches. Model fits further show that a single-rate process outperformed a dual-rate process. We suggest that a single-rate process underlies Coriolis force reach adaptation, perhaps because these forces relate to familiar body dynamics and are assigned to an internal cause.

## Introduction

We have the ability to learn from our mistakes. In motor learning, one of the types of mistakes is the difference between the actually sensed and the internally predicted sensory consequences of a movement. These so-called sensory prediction errors, which could arise from internal sources (e.g. execution noise, sensory bias, muscle fatigue) or external sources (e.g., unforeseen forces on the body), are used to adapt future movement plans [1].

It has been suggested that sensory prediction errors drive multiple adaptive processes, with some adapting and forgetting quickly while others adapt more slowly but retaining for longer

**Funding:** This work was supported by a grant of the Swedish Research Council (JCS, grant nr. 2016-01725) and an internal grant from the Donders Centre for Cognition (WPM). The funders had no role in study design, data collection and analysis, decision to publish, or preparation of the manuscript.

**Competing interests:** The authors have declared that no competing interests exist.

[2–6]. Together these processes support the system to quickly adapt to abrupt perturbations but also to slowly change behavior more permanently to persistent perturbations [5].

Evidence for multiple adaptive processes comes primarily from paradigms of spontaneous recovery, which demonstrate a rebound effect of the adaptation to an initial long exposed perturbation (say A), after it was followed by a brief reverse-adaptation to the opposite perturbation (say B). To explain this rebound, Smith and colleagues (2006) proposed a dual-rate adaptation model, with a fast and slow state, in which the brief adaptation towards B is driven by the fast adaptive state, whereas the subsequent re-expression of the adaptation to A is caused by the lagging slow state that has not yet transitioned to compensate for B [5]. Support for dual-rate learning has now been reported for adaptation of reaches in force fields [5–7] and under visuomotor perturbations [2,3,8] as well as for saccadic gain adaptation [9,10]. Neural signatures of dual- and multi-rate adaptive processes have recently been shown [4,11].

However, not all adaptive motor behaviors are consistent with a multi-rate model of adaptation. For example, Ingram and colleagues (2011) have shown that adaptation to the dynamics of a familiar object, such as a hammer, is better explained by a single-rate than dual-rate adaptation model [12]. The authors suggested that the familiarity with the hammer-like object dynamics make the adaptation process fundamentally different from adaptation to unfamiliar mappings and forces, like visuomotor rotations and curl-force fields.

Another type of familiar dynamics are Coriolis forces, which are contact-free forces on the arm that arise when we reach for an object while our torso is rotating [13]. If the torso is actively turned, the brain is able to predict and compensate for the ensuing Coriolis forces in the ongoing reach. However, also during passive turns, the brain is known to adapt to these familiar forces within a few repeated trials [13]. Here we ask the question whether this rapid adaptation is mediated by a single-rate process, analogously to the rapid adaptation to hammer dynamics, or governed by a multi-rate learning process.

Human participants were placed in the center of a fully-enclosed rotating room, spinning at constant speed, and instructed to make alternating forward and backward reaching movements between two body-fixed targets. There was first a long phase with ample trials for the reaches to adapt to the evoked Coriolis forces based on visual endpoint feedback. Next, the rotation direction of the room was reversed, and participants re-adapted their reaches for a few trials to the reversed Coriolis forces. Subsequently, while the room was stationary, reaches were made without visual feedback. We hypothesized that if the adaptation was governed by a multi-rate learning process, reach compensation for the first rotation (spontaneous recovery) would be observed during this phase.

Because participants made contact-free movements in all phases, we cannot measure the ideal force compensation, as is typically done in robotic force adaptation experiments. Here, we therefore assessed the movement trajectory–the hand path error–as an index of adaptation, and fit a single-rate and dual-rate adaptation model to the individual participant data. Our behavioral results were more parsimoniously explained by the single-rate model. We suggest that a single-rate learning process mediates Coriolis force adaptation, perhaps because these forces mimic exposure to familiar body dynamics.

## Materials and methods

### Participants

Our study involved 34 right-handed participants (15 female), without any known balance problems, inner ear abnormalities or history of motion sickness. Their mean age was 24.8 years ($SD$ = 4.0). Data of seven participants were excluded from analyses due to technical problems (five participants) or failure to follow task instructions (moving their arm while the room

accelerated or during block B). Of the 27 remaining participants, 17 formed the experimental group and 10 participants formed the control group. The study was approved by the ethics committee (approval code: ECSW-2018-083) of the Social Sciences Faculty at Radboud University Nijmegen, The Netherlands. Participants gave written informed consent prior to their participation.

### Experimental setup

Participants were seated in fully enclosed *rotating room* (Fig 1A). The room had an octagon-shaped layout with a radius of 1.45 m and a height of 2 m and could rotate in the horizontal plane. The total mass of the room (including participant and experimenter, who was also in the room) was about 1250 kg. Supported by 12 running wheels at the outer edges, and a swivel in its center, it was powered by a brushless servomotor motor (MotorPowerCo, type: T142.16.5.15.6.E1.0.G2, tetra & tetra compact). With an acceleration of 1 rpm$^2$, it took 6 s to achieve an angular speed of 10 revolutions per minute (rpm). Participants were seated in the center of the room with their head fixated such that the rotation axis of the room was between

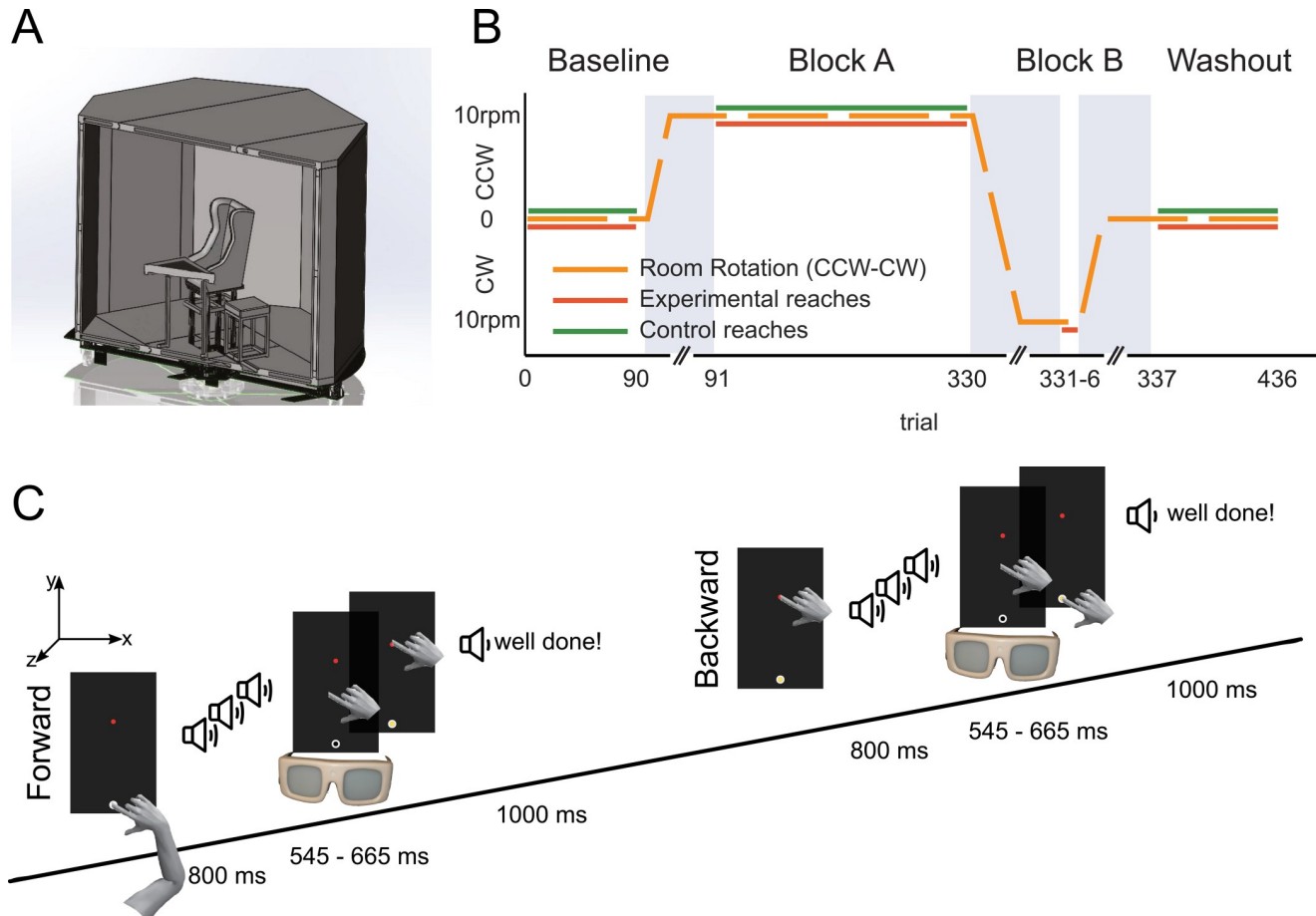

**Fig 1. Experimental setup, paradigm and task.** (A) Participants were seated in the center of a fully-enclosed rotating room and performed right-hand reaching movements to visual targets presented on a touch screen. (B) Experimental paradigm. Dark orange and green solid lines respectively indicate during which phases the experimental and control group made reaches. The dashed orange line indicates the rotation speed of the room. Grey areas indicate acceleration and deceleration of the rotating room and the time to let canal effects dissipate. (C) Temporal sequence of one trial pair (forward and backward movement). Reach is instructed by three auditory cues; vision is blocked during the reach.

their right shoulder and the center of their head. The experimenter sat next to the participant and communicated via an audio and via a visual channel to an assistant outside the room controlling the room's motion.

An Iiyama touch screen monitor (59.8x33.6 cm, 1920x1080 pixels i.e. 27-inch, ProLite; Iiyama, Tokyo, Japan) was positioned horizontally in front of the participant at chest level (in portrait, with the screen facing up). Participants were instructed to make forward (FW) and backward (BW) reaching movements, between two visually presented targets (yellow and red circles, diameter 2 cm) at a mutual distance of 35 cm, in a sagittal plane midway the sternum and right shoulder. The yellow target was closest to the body and was approximately 15 cm from the chest. Participants wore shutter-glasses (PLATO Visual Occlusion Spectacles, Translucent Technologies Inc. Toronto, Canada), which closed at the onset and opened again at the offset of a reach (Fig 1C), determined ideally based on contact of the hand with the touch screen but in a number of subjects was based on experimental timings to create a smoother pace of the experiment (see below for further details).

The target closest to the body was encircled by tape (3M Transpore white surgical tape), so participants could also use touch to locate it prior to each forward movement (Fig 1C). Three sequential beeps were used as ready-set-go signal for a movement, each 400 ms apart. To differentiate between forward and backward trials the pitch of the first beep was 390 Hz for forward, and 490 Hz for backwards movements. The pitch of the second, 390 Hz, and third beep, 440 Hz, were identical for forward and backward trials. After each trial, participants received auditory feedback about their movement speed, i.e. the time between touchscreen release and retouch. If their movement took longer than 665ms, a pre-recorded voice instructed them to *'move faster'*, if their reach took less than 545ms, the instruction was to *'move slower'*, and in the remaining cases participants were told that they had done well (*'well done'*). Because every movement contributes to the learning, irrespective of their durations, we did not reject trials based on these criteria. The intertrial interval was 1 s (including feedback), so the total duration of one trial was approximately 2.4 s (i.e. sum of beep time: 0.8 s, Movement Time: 0.6 s, and ITI: 1 s).

In the initial series of experimental sessions, the shutters did not close based on hand release of the touch screen in about 3.50–6.20% of the trials. Note that we have included all trials, even those in which the shutters did not close. Although this is only a small percentage, several participants indicated frustration because it affected the pace of the experiment. We resolved this in the remaining sessions (all control participants and three participants of the experimental group) by controlling the shutters based on timers (close at the *'go'*-signal, i.e. third beep, and open 605 ms later, i.e. at the instructed movement time) rather than touchscreen events.

The 3D location of spherical motion tracking markers (diameter of 7 mm) were recorded at 100 Hz using a room-fixed motion tracking system (Qualisys, Miqus M5 camera system) containing 7 motion tracking cameras. In addition, a synchronized and position calibrated video camera (Miqus color) recorded the participant at 25 Hz. The motion tracking markers were attached to the tip of the index finger, the elbow joint, and shoulder joint of the participant's right arm. Furthermore, two rigid bodies, each equipped with 4 markers, were attached to upper and lower arm to track their orientation. Of those, only the data of the index finger marker was used in the present analysis (see below). Furthermore, three additional markers were attached to the corners of the touch screen and were used to re-define the Cartesian axis system within the rotating room and align the workspace across participants (screen moved between sessions to seat participant, this way start and end points were always aligned). Presentation software (Version 18.0, Neurobehavioral Systems, Inc., Berkeley, CA) was used to present stimuli, detect touch events, send triggers to the Qualisys system, and control the shutter-glasses.

## Paradigm

Prior to the main experiment, participants were familiarized with the task in at least 90 trials, training them on the movement timings and on reaching under the use of shutter glasses, up until they were comfortable with the task. The main experimental paradigm consisted of 4 blocks (Fig 1B): a baseline reach block (90 trials) where the room was stationary, a long reach adaptation block while the room rotated (block A, 240 trials), a short reach adaptation block while the room rotated in the opposite direction (block B, 6 trials), and a reach washout block (100 trials). A trial pair is defined as a forward and backward movement. Hence, there were 45 trial pairs in the baseline block, 120 trial pairs in block A, 3 trial pairs in block B and 50 trial pairs in the washout block. Participants were randomly assigned to either of two groups: one group started with clockwise (CW) rotation of the room, the other started with counterclockwise (CCW) rotation. Seven participants from the experimental group and six participants from the control group were tested with CW rotation in block A and CCW rotation in block B. Ten participants of the experimental group and four participants from the control group were tested in the opposite order.

During the baseline block, participants had a short break of 10 s after every 30 trials, during which the hand rested at the right side of the touch screen. After the baseline block, the room was brought to a constant speed of 10 rpm to either CW or CCW rotation. After the room had reached the required constant rotation speed, and an additional 30s break with the lights on had passed (used to let canal effects dissipate, [14]) the lights were dimmed and the participant made the 240 reaches, with intervening breaks of 10 s after every 30 trials. Subsequently, the room was decelerated and accelerated (12 s of 1 rpm$^2$) to reach a constant rotation speed (10 rpm) in the opposite direction (Fig 1B). Next, again after a 30 s break with the lights turned on, participants in the experimental group made six reaches, while the control participants made no reaches during the same time period. Finally, the room decelerated for 6s to come to a standstill. After another 30 s break with the lights turned on, the washout block started during which participants completed 100 reaches without breaks.

In order to emulate the error clamps that are used in spontaneous recovery paradigms [5], visual feedback was only provided at the start of the forward movement during the wash-out block to allow participants to see where to move, but to exclude visual feedback about the end location of their reach. This meant that the shutters were only opened when the participants index finger was at the start position, which could be located by a tactile cue (see above). Visual feedback about the start position of the backward movement was inadvertently available for three control participants during their washout blocks. The respective trials were included in the analyses, since they did not systematically deviate from the data of the other participants.

## Data-analysis

Qualisys Track Manager (Version 2018.1) was used to identify the 3D positions of the finger and touchscreen markers. Timing information of the start and end of the different blocks of the paradigm (baseline, Block A, Block B and wash-out) were added manually to each participant's data-set (based on video data). Marker position data were further analyzed in MATLAB (2017b). All data were expressed in a Cartesian coordinate system, first based on the calibration of the Qualisys system and later aligned to the markers on the touchscreen. The y-axis was pointing forward, parallel to the mid-sagittal plane of the participant, the x-axis pointing rightward, and the z-axis pointing up from the screen. The origin of the coordinate system (after alignment) was defined as the start position of the forward movement. Data were segmented into trials based on the y-position of the index finger data (alternating between positions closest to the two target marker positions). Cubic spline interpolation was used to replace missing values. Marker position data were filtered using a fifth-order, 12 Hz low-pass bidirectional

Butterworth filter, before 3D marker velocity was calculated by taking the central difference for each time step. Marker speed was taken as the norm of 3D marker velocity. For each trial, the onset and offset of the reach were defined as the times at which finger marker speed first exceeded 10 cm/s and when, after peak speed, crossed this boundary again. Trials without a clear velocity peak (i.e. speed $\leq$ 35cm/s) as well as trials with a 2D (x, y coordinates) movement amplitude outside the 22–52.5 cm range (targets were 35 cm apart) were excluded (on average 2.7 trials). For further analyses of the remaining trials (95.1%), we only considered data (position and velocity) in the horizontal plane (x, y coordinates). Trial data were resampled using linear interpolation, such that every movement was time-normalized and contained 100 samples.

**Trajectories.**   The time-normalized position data were used to calculate the mean and standard deviation of the reach trajectory across participants for each trial of the paradigm. The grand mean across trials was computed for the baseline block. For visualization purposes, data of experimental and control participants were combined in the baseline and in block A (since the paradigm does not differ in these blocks for the two groups).

**Lateral deviation at maximum speed.**   As a measure of the kinematic error on a single trial, we used the lateral deviation (LD) at maximum speed of the trajectory relative to a line through the start position of the movement, parallel to the y-axis. The LDs of a forward and backward movement pair were subsequently averaged. In addition, we investigated the endpoint error, the maximum absolute error (again relative to the line through the start position, parallel to the y-axis) and the maximum perpendicular error (to a straight line, from start to end of the trajectory). Except for the endpoint error, all error measures showed similar patterns as compared to the LD although some measures were more noisy. The end-point error mainly differed in pattern from the LD in the error magnitude in block B, which did not exceed the magnitude of the error in block A. This is also visible in the trajectories (Fig 2), in which the error correction at the end of the movement is larger in the second rotation direction as compared to the first rotation direction. We assume this is caused by increased visual and proprioceptive feedback gains over the course of the experiment [15,16], that especially affect the final part of the movement.

## State-space modeling

To interpret the adaptation patterns, we fit a single- and dual-rate adaptation model to the individual participant's LD data. We will first explain the details of the two models, followed by the fitting procedure and statistical methods to compare the two models.

**Dual-rate model.**   The dual-rate model, as proposed by Smith and colleagues (2006) [5], specifies fast and slow states, each of which depends on the estimated perturbation state at the previous trial, multiplied by a retention factor, and the prediction error of the current trial multiplied by the learning rate. The single-rate model is the simpler version, containing only a single state. As described in more detail below, we adjusted both models for inter-block memory decay, as well as the reduced error feedback in the washout block.

For both models, the prediction error is computed as,

$$e(t) = p(t) - x(t) \tag{1}$$

where $e(t)$ is the prediction error on trial $t$, i.e., the difference between the predicted perturbation $x(t)$ and the actual perturbation $p(t)$ on that trial. As a proxy for the actual perturbation caused by the CW room rotation in block A, we took the observed trajectory deviation (LD) on the first trial:

$$p(t_A) = LD(A_1) \tag{2}$$

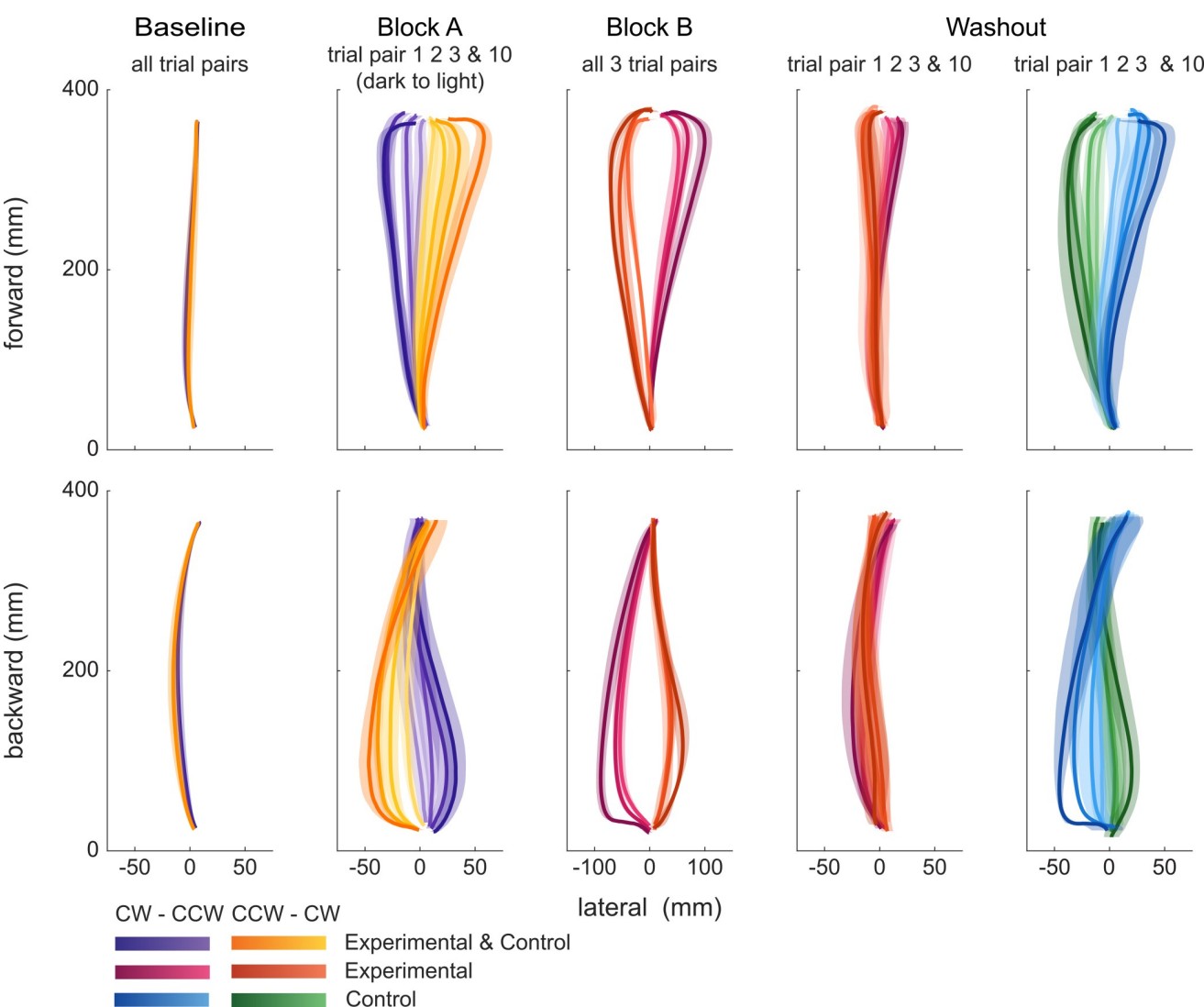

**Fig 2. Reach trajectories at different phases of the paradigm.** CW-CCW groups, cooler colors (i.e. purple, magenta and blue); CCW-CW group; warmer colors (i.e. ochre, orange and green). Top row, forward reaches; bottom row, backward reaches. Baseline: grand-average of all trials of control and experimental group. Block A: Across-participant average of forward and backward reaches in trial pair 1, 2, 3, and 10 of the block (i.e. trial pairs 46, 47, 48, and 55 of the paradigm), ordered from darker to lighter hues. Block B: Across-participant average of each of the three forward and backward reaches, ordered from darker to lighter hues. Washout: Across-participant average of forward and backward reaches in trial pair 1, 2, 3, and 10 of the block, in separate panels for the control and experimental group.

Furthermore, the perturbation magnitude for the opposite room rotation in B was defined as a scaled version of the perturbation magnitude in A:

$$p(t_B) = cp(t_A) \tag{3}$$

in which $c$ was a free parameter to allow that the Coriolis force had a different perturbation magnitude for the two rotation directions as a result of differences in arm dynamics.

The estimate of the perturbation magnitude on a trial depends on the update of two state estimates:

$$\vec{x}(t+1) = R\,\vec{x}(t) + \vec{l}\,e(t) \tag{4}$$

where the vector $\vec{x}(t+1)$ entails the fast and the slow states after updating: $\begin{bmatrix} x_f \\ x_s \end{bmatrix}$. $R$ is a matrix containing the retention factors, $\begin{bmatrix} r_f & 0 \\ 0 & r_s \end{bmatrix}$ with $r_f$ the retention factor of the fast state and $r_s$ the retention factor of the slow state, and $\vec{l}$ is a vector containing the learning rates, $\begin{bmatrix} l_f \\ l_s \end{bmatrix}$ where $l_f$ is the learning rate of the fast process and $l_s$ is the learning rate of the slow process. The learning and retention factors were constrained as follows: $r_f < r_s < 1$ and $l_f > l_s > 0$ [5]. The estimate for the perturbation on a trial $x(t)$ is the sum of the two state estimates:

$$x(t) = \begin{bmatrix} 1 & 1 \end{bmatrix} x(t) \tag{5}$$

**Single-rate model.** According to the single-rate model, adaptation takes place by updating a single state. As in the dual-rate model, the prediction error is defined as the difference between the expected perturbation on that trial and the actual, perceived perturbation on that trial, as described in Eq 1. The perturbation magnitude was quantified by the naïve error as observed when reaching for the first time in the rotating room (see Eqs 2 and 3).

The estimated perturbation for the next trial depends on the estimate for the current trial, and the error observed in the current trial. How much the error contributes to the perturbation estimate for the next trial depends on the learning rate $l$. The amount of retention of the previous trial is defined by the retention factor $r$:

$$x(t+1) = rx(t) + le(t) \tag{6}$$

For a single-rate model, $r$ and $l$ are both scalars.

**Washout phase.** Our paradigm involved no error clamps during the washout, and hence did not exclude error feedback during this block as typically done in robotic force field studies. Although we excluded all visual feedback, participants could still use proprioceptive feedback from their compensatory movements to drive the adaptation during the washout. To account for this reduced error feedback, we extended both models with an additional parameter ($l_{wo}$) to scale down the (un-)learning during the washout block. This results in the following dynamics for the washout block in the dual-rate model:

$$\vec{x}(t+1) = R\vec{x}(t) + l_{wo}\vec{l}\,e(t) \tag{7}$$

and for the single-rate model:

$$x(t+1) = rx(t) + l_{wo}le(t) \tag{8}$$

**Breaks.** Finally, we extended our state-space models with time-dependent decay (break parameter b) to account for the long transition phases between the rotation directions of the room [8]. The break parameter b specifies how many repeats of trial-to-trial decay (1-retention factor) would have occurred if there were no breaks. The between block break was 1.5 times longer between block A and B, then between block B and washout, which was accounted for by an additional scaling factor $d$ [8].

Thus, in both models, the state of the first trial after a break depended on the last trial before the break and the elapsed time. For the dual-rate model this results in:

$$\vec{x}(t+1) = R_{break}\vec{x}(t) + \vec{l}\,e(t) \tag{9}$$

with

$$R_{break} = \begin{bmatrix} r_f^{db} & 0 \\ 0 & r_s^{db} \end{bmatrix} \qquad (10)$$

And for the single-rate model:

$$x(t+1) = r^{db}x(t) + le(t) \qquad (11)$$

The scaling factor $d$ was set to 1.5 for the inter-block break between A and B and to 1.0 otherwise.

**Model fitting.** Values for the parameters of the single-rate and dual-rate model were estimated at the individual participant level by minimizing the mean squared difference between the model prediction and the observed *LD*. This fitting procedure was applied 200 times, using Matlab's fmincon function, with randomly selected starting values for the parameters. We selected the set of best-fit parameters that yielded the smallest mean squared error. Bounds on the parameters were as follows: for all learning ($l$, $l_s$, $l_f$) and retention ($r$, $r_s$, $r_f$) rates the lower bounds were set to $10^{-4}$ and the upper bounds were set to 0.999. For the dual-rate model, additional constraints were $ls - lf \leq -10^4$ and $-rs \leq -10^4$. The lower bound of the washout learning parameter, $a_{wo}$, was set to zero and the upper bound to one. The perturbation scaling parameter $c$ for block B was bound between 0.5 and 1.5. The $b$ for the memory decay in the breaks was constrained between 0 and 15.

**Model comparison.** Model fits of the single- and dual-rate model were compared by calculating the *Bayesian information criterion (BIC)*, following Berniker and colleagues [17]: *BIC* = $n\,ln(MSE) + k\,ln(n)$. The BIC corrects for the greater number of parameters ($k$) in the dual- compared to the single-rate model (dual-rate: 7 free parameters; single-rate: 5 free parameters). The number of observations ($n$, the number of trial pairs) was between 196 and 218 (depending on the missing trials). A BIC difference larger than 6 is considered to provide strong evidence and a BIC above 10 is considered to provide very strong evidence for one of the models [18].

## Results

Participants made reaches away and toward the body in a rotating room, following a spontaneous recovery paradigm consisting of four phases. After an initial baseline period in which the room was stationary, there was a long reach adaptation phase (Block A) in which the room rotated at constant velocity. Subsequently, the rotation direction was reversed and participants were exposed to this new dynamics for a few reaches (Block B). Finally, the room rotation ceased and participants made reaches while the room was stationary (washout block).We tested the emergence of spontaneous recovery effects during this washout period, to examine whether reach adaptation to Coriolis forces is best described by a single-rate or dual-rate learning process using a model-based analysis. For comparison to each experimental group, we tested a control group using the same protocol but without making reaches in block B, for which we expect no rebound effects to emerge during the washout phase.

Fig 2 illustrates the mean time-normalized trajectories ($\pm$ *SE*, shaded regions) of the four groups (two experimental, two control) during the various phases of the experiment. Because participants in the experimental and associated control group performed the same trial series during the baseline block and block A, grand-averages per combined group are presented. While slight differences in curvature can be seen between the forward and backward reach trajectories, there are no differences between the groups during the baseline block, as expected.

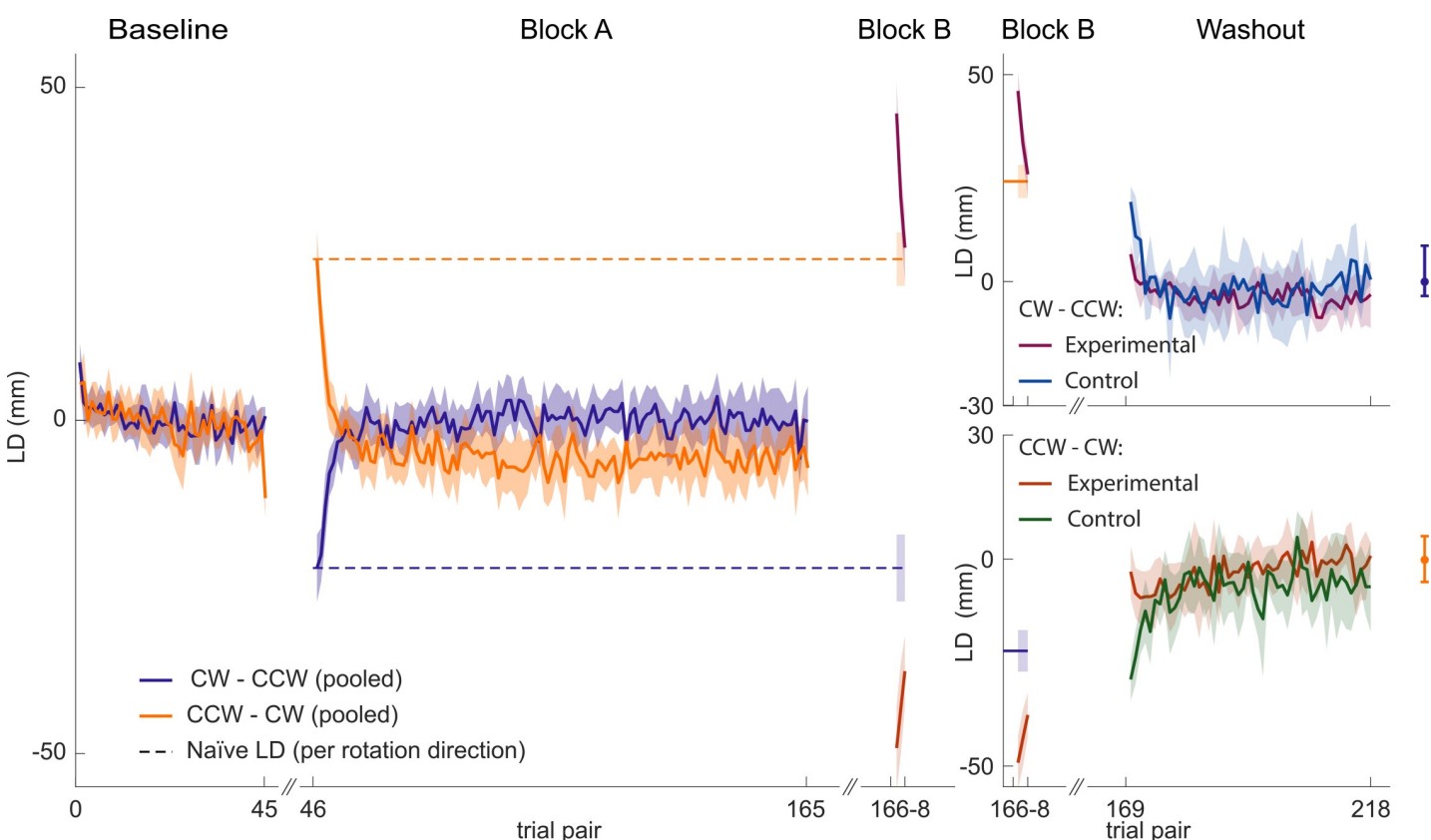

**Fig 3. Lateral deviation as a function of trial pair number.** CW-CCW (purple, pink and blue lines) and CCW-CW (orange, dark orange and green lines) group. Baseline and block A: grand-average of all trials of control and experimental group (orange and purple). Block B and washout: control (green and blue) and experimental group (pink and dark orange) plotted separately. Baseline fluctuations are indicated to the right of the washout phase by the mean, maximum and minimum of the mean baseline levels.

During block A, the participants' initial reaches show large deviations from the baseline reaches. These deviations are in opposite directions for the CW (purple traces) and CCW (orange traces) rotation groups. Because participants experience opposite Coriolis forces for the forward and backward reaches, trajectories also deviate in opposite directions. In both cases, trajectories show the highest curvature towards the end of the movement, where presumably feedback mechanisms kick in to achieve the target. Over repeated trials, trajectories become straighter–a marker of adaptation–and after ∼ 10 trial pairs, the reach trajectories (lightest shades) cannot be distinguished anymore from the baseline trajectories. Participants performed 110 more trial pairs while being exposed to the same Coriolis forces, but virtually no further changes are visible (see Fig 3).

In block B, during which room rotation was reversed, only the two experimental groups made reaching movements, whereas the control groups kept their arm still while also being exposed to the rotation. Initial reaches in block B show much larger deviations and stronger curvature (darkest colors) than the initial reaches in block A (note that the abscissae have different scales). This observation can be explained by an active compensation for the expected force learnt in block A, while the experienced and uncompensated Coriolis force is now in the same direction as the force compensation. Over the 3 forward and 3 backward reaches in block B, all of which are shown, the trajectories show clear but incomplete adaptation. More specifically, the trajectory of the sixth trial is still quite different from baseline, as if the previous

compensation for the Coriolis force in block A has been partially un-learned, but the induced force in block B still has to be learned.

In the washout block clear differences between the experimental and control group appear. Both control groups show clear aftereffects of compensating for the forces experienced in block A, suggesting that the exposure to the opposite rotation in block B has not hampered retention or induced adaptation. On the other hand, both experimental groups show much smaller trajectory deviations and look more similar to the baseline trajectories. The question is, do these trajectories demonstrate effects of spontaneous recovery, i.e., a rebound toward the compensation for the Coriolis forces experienced in block A? If so, this would be evidence for a dual-rate learning process. According to the dual-rate adaptation model, the fast state already learned to compensate for a large part of the perturbation in block B, but the slow state still lingers in compensating for the perturbation in block A. The observed behavior is the summed effect of both states, explaining the partial compensation in block B. In the washout phase, where error-based adaptation is diminished, the fast state quickly forgets the compensation learned in block B while the slow state still retains the compensation for block A. This should then cause a gradual re-emergence of compensation for block A that subsequently slowly reduces to no compensation at all.

Alternatively, if there are no signs of spontaneous recovery of the adaptation to the perturbation experienced in block A in the washout block, this hints at a single-rate learning process. Indeed, from a single-rate model, lateral deviations of trajectories in the washout block should converge to baseline from the compensation levels present in the final trial of block B, without any rebound. So if final compensation was still for the rotation in block A, compensation should washout to baseline without crossing it. Similarly, if final compensation was in the direction of block B, it should washout to baseline without ever showing compensation for block A.

From the raw trajectories it is impossible to discern whether they are a reflection of a single-rate or dual-rate adaptation process. Therefore we summarized the reach trajectories of each forward-backward trial pair into a single number, the lateral deviation at maximum speed (LD, see Materials and Methods). Fig 3 shows how the LD evolves as a function of block and trial pair, averaged across participants. The LD straddles around zero during the baseline trials, consistent with the relatively straight trajectories in Fig 2. At the start of block A, the LD initially clearly deviates from zero, but then quickly returns to zero or even slightly crosses zero (CCW-CW group) towards a level opposite to the initial deviation (indicating overcompensation) in $\sim 10$ trial pairs.

The LD during block B (trial pair 166–168) characterizes the trajectories after the room has switched rotation direction. The dashed lines indicate the predicted initial deviation had participants been exposed to this rotation after the baseline block (as taken from the other rotation group's initial performance during block A). As shown, the initial LD in block B exceeds that value (due to the learned compensation from block A and the uncompensated Coriolis force experienced in block B). However, the LD quickly approaches the naïve value in the subsequent trial pairs, particularly during the reaches of the CW group (purple). The CCW group (orange) did not fully unlearn the adaptation to the forces of block A.

During the washout block, starting at trial pair 169 and containing 50 pairs of forward and backward reaches, the control groups show clear aftereffects of block A, despite having experienced the opposite rotation in block B without making reaches. The aftereffects of block A quickly fade away in about 10 trial pairs (blue: CW control group, green: CCW control group).

As stated earlier, the CW experimental group seems to have fully unlearned the compensation for the rotation in block A at the end of block B. This means that any aftereffect consistent with the rotation in block A during the washout block would be indicative of a dual-rate

learning process. However, even though the observed aftereffect (magenta, top-right panel) has the same sign as the aftereffect of the corresponding control group (blue, top-right panel), it does not show the pattern typically observed in spontaneous recovery. That is, we do not observe that the compensation to the initial perturbation quickly rises (due to the forgetting of the fast process) up to maximal recovery and then slowly decays due to forgetting of the slower process. The response of the CCW experimental group during the washout trials (orange) shows the typical rise-and-fall pattern, but because this group did not unlearn the compensation of block A during block B, its interpretation is ambiguous. Also, in both groups the aftereffects do not exceed the observed fluctuations in the baseline.

To infer whether a single- and dual-rate process governed the patterns in our behavioral data, we fitted these two models to the individual participant LD data. Fig 4 illustrates the behaviorally observed LD and the respective model fits of three representative participants. The left panel of the top row shows the data of a participant for which there are no spontaneous recovery effects discernible in the behavioral data and the best-fit lines are overlapping for the two models (single-rate: orange-dashed line; dual-rate: green line). The two-middle panels, expanding on the trials for which the models are supposed to differ most in their predictions (the initial adaptation in block A and B, and the initial washout phase), also show no difference. This suggests that the single-rate model provides a more parsimonious explanation of the data, which is confirmed by a BIC analysis (BIC single-rate = 896.48; BIC dual-rate = 907.25). This is further illustrated in the right-hand panel showing that the prediction of the dual-rate model is driven by only one of its states. This pattern was observed in five other participants as well (see S2 Fig). The middle row shows the data and model-fits of another participant. The participant's LD data resembles the pattern of the participant in the top row, including the absence of a clear spontaneous recovery effect. Fitting the two models revealed again overlapping curves (see also the middle panels), explaining why a BIC analysis favored the single-rate model (BIC single-rate = 738.17; BIC dual-rate = 748.98). Analysis of the dual-rate model revealed that both states converged to the inequality constraint (i.e. converged to the same learning rate and retention factor values for the fast and the slow state), which is just another way to approximate a single-rate model. A similar pattern was found in six other participants (see S3 Fig). The bottom row of Fig 4 illustrates a participant whose LD data may show some signs of spontaneous recovery, i.e. data points in the washout period transition from below to above the zero LD line. While the best fits for the dual- and single-rate model seem to overlap, closer scrutiny (middle panels) shows that the dual-rate model predicts steeper initial learning during block A than the single-rate model, as well as a small hint of spontaneous recovery during the washout block. If we look at the slow and fast state of the dual-rate model (right panel), they indeed show different dynamics. Yet, a BIC analysis suggests that the single-rate model is still a more parsimonious explanation of the data, despite the data containing hints of a dual-rate model (BIC single-rate = 828.39; BIC dual-rate = 833.22). A similar pattern was found in three other participants (see S1 Fig) but in none was the BIC in favor of the dual-rate model.

So, both models can account reasonably well for the systematic variation of the data. As an indication of the quality of the model fits, we compute the $R^2$ to the fit and data of the second learning (B) and spontaneous recovery block. This value ranged between 0.32 and 0.80 among participants, and had a mean of 0.64 (SD = -.12) across both models and all participants. Examining the variance of the residuals, the two models show only minute differences across participants (single-rate: $MSE$ = 55.1 mm, $SD$ = 26.0 mm; dual-rate: $MSE$ = 54.9 mm, $SD$ = 25.8 mm). But, as shown in Fig 5, BIC values show strong ($\Delta BIC > 6$) to very strong ($\Delta BIC > 10$) evidence in favor of the single-rate model over the dual-rate model in all but one of our participants (participant 9 see Fig 4). In this one participant the evidence in favor of the

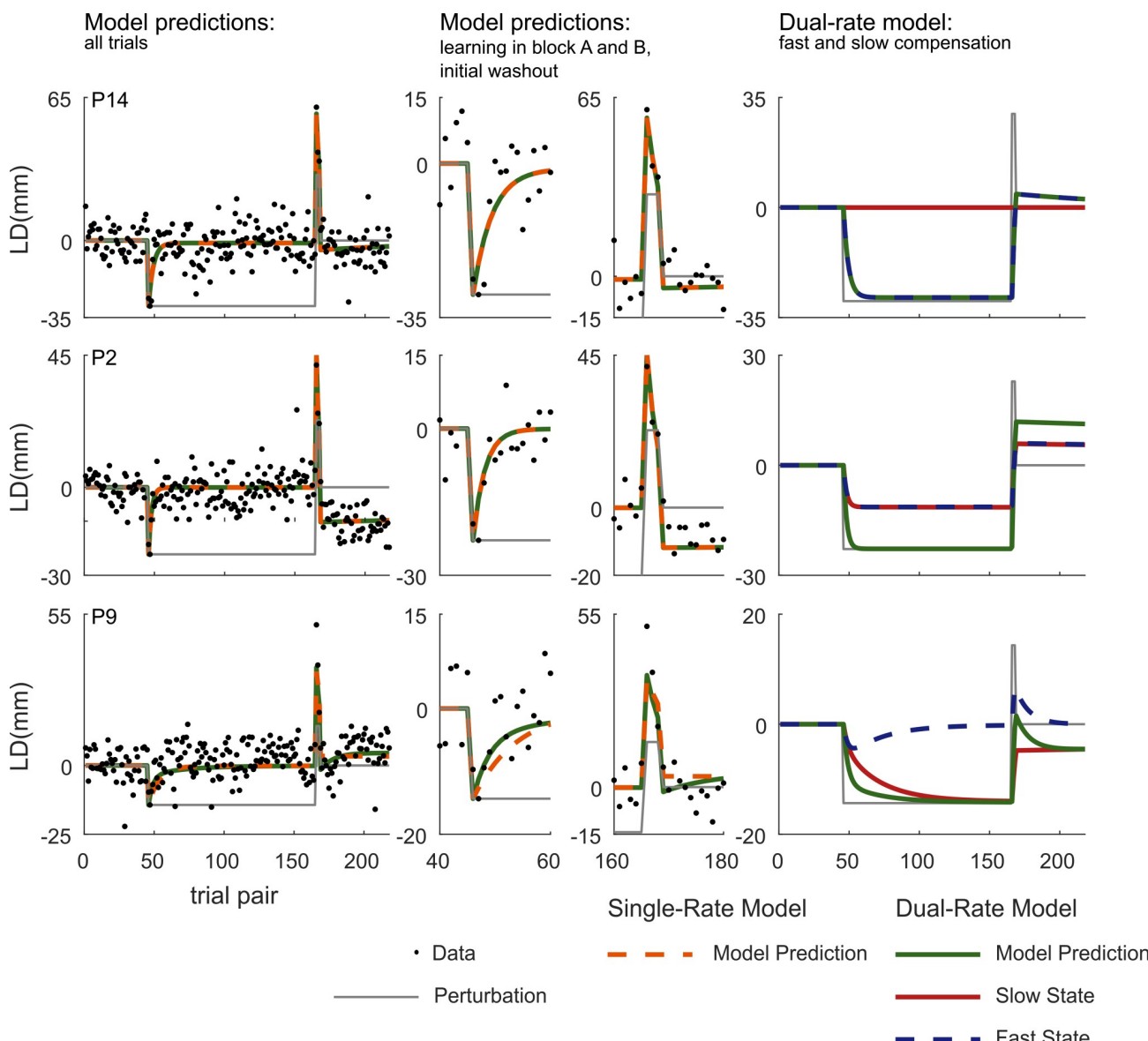

**Fig 4. Data and model fits of three representative participants.** Left panels: Model prediction of a single- (orange) and dual-rate (green) model per participant (individual rows), including the perturbation scheme (gray) and the data (black dots). Middle panels provide a detailed view of early learning in A, B and the washout. Right-panels: contributions of the fast and slow state to the dual-rate model fit.

single-rate model over the dual-rate model was only weak ($\Delta BIC = 4.83$). Table 1 lists the values of the best-fit parameters of the two models. As described above, the parameters of the dual-rate model follow three different patterns of which two mimicked features of a single rate model. For the single-rate model, the learning rates vary between 0.12 and 0.48 with a mean learning rate of 0.24 ($SD = 0.09$) across participants. The retention factors ranged from 0.976 and 0.999 (i.e. the upper bound) with a mean retention factor of 0.993 ($SD = 0.01$).

## Discussion

We investigated whether reach adaptation to Coriolis forces due to passive whole-body rotation is governed by a single- or dual-rate learning process. We utilized a paradigm that is

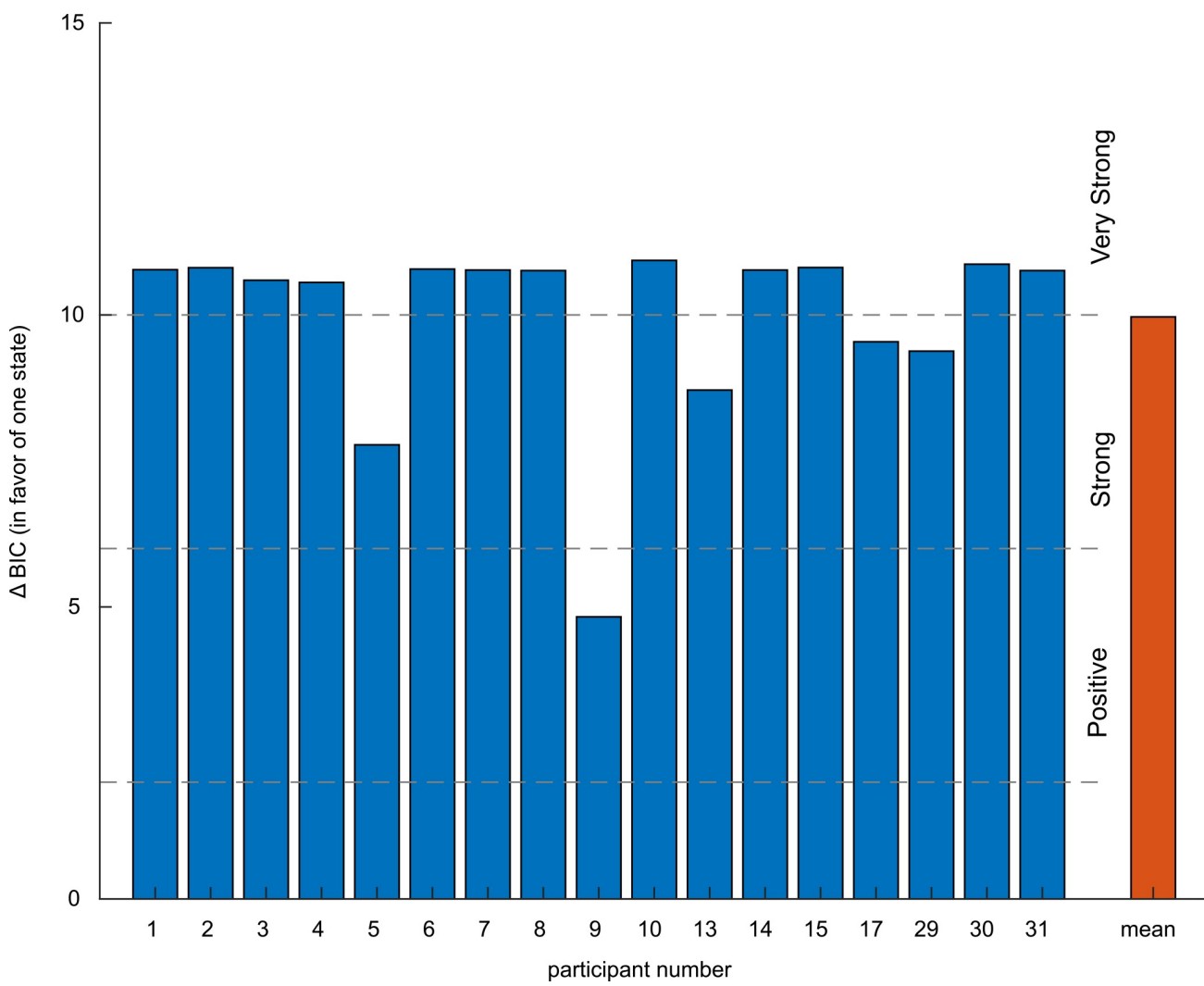

**Fig 5. Difference in BIC per participant.** BIC of the single-rate model subtracted from the BIC of the dual-rate model.

known to evoke spontaneous recovery under the operation of a dual-rate, but not a single-rate process. Participants made a substantial number of reaches under one Coriolis force perturbation followed by a few reaches under the opposite perturbation (opposite room rotation). We examined whether the reach kinematics over trials during the washout phase (without perturbation) showed spontaneous recovery by transitioning from compensating for the preceding Coriolis force to compensating for the force experienced during the first rotation block. Compared to the control group, participants in the experimental group show only small aftereffects of the first rotation in the washout phase and no clear signs of spontaneous recovery. Model fitting and comparison revealed a more parsimonious account of the data by the single-rate than dual-rate model in each of the individual participants.

The suggestion that a single-rate learning process underlies adaptation to Coriolis forces contradicts findings in most other motor adaptation paradigms, that point toward a dual- or multi-rate learning process [2,3,5,10]. The advantage of a dual/multi-rate process is clear: it enables flexible adaptation to sudden perturbations but also enables more long lasting changes to persistent perturbations [5]. Although the present results suggest a single-rate model for

**Table 1. Parameters of the single-rate and dual-rate model fit for individual participants of the experimental group.**

| participant | model single dual | learning rate(s) l $l_f$ | $l_s$ | retention factors(s) r $r_f$ | $r_s$ | $l_{wo}$ | c | b | Δ BIC |
|---|---|---|---|---|---|---|---|---|---|
| 1 (CW) | single | 0.2678 | | 0.9984 | | 0 | 0.85 | 15 | 10.77 |
| | dual | 0.2501 | 0.0175 | 0.9983 | 0.9985 | 0 | 0.85 | 14.97 | |
| 2(CW) | single | 0.3738 | | 0.999 | | 0 | 1.01 | 15 | 10.81 |
| | dual | 0.1869 | 0.1868 | 0.9989 | 0.999 | 0 | 1.02 | 15 | |
| 3 (CCW) | single | 0.264 | | 0.999 | | 0 | 0.76 | 15 | 10.59 |
| | dual | 0.132 | 0.1318 | 0.9989 | 0.999 | 0 | 0.76 | 15 | |
| 4 (CW) | single | 0.2157 | | 0.9786 | | 0 | 1.28 | 15 | 10.56 |
| | dual | 0.2156 | 0.0001 | 0.9786 | 0.9787 | 0 | 1.28 | 15 | |
| 5 (CCW) | single | 0.1669 | | 0.999 | | 0 | 0.5 | 0 | 7.780 |
| | dual | 0.2001 | 0.192 | 0.66 | 0.999 | 0 | 0.5 | 3.94 | |
| 6 (CW) | single | 0.4785 | | 0.999 | | 0 | 0.52 | 0 | 10.78 |
| | dual | 0.2394 | 0.2393 | 0.9989 | 0.999 | 0 | 0.52 | 0 | |
| 7 (CW) | single | 0.2585 | | 0.9754 | | 0 | 0.77 | 7.59 | 10.77 |
| | dual | 0.2583 | 0.0002 | 0.9754 | 0.9755 | 0 | 0.77 | 7.59 | |
| 8 (CW) | single | 0.1471 | | 0.9663 | | 0 | 1.5 | 0 | 10.76 |
| | dual | 0.1459 | 0.0012 | 0.9663 | 0.9665 | 0 | 1.5 | 0 | |
| 9 (CW) | single | 0.1186 | | 0.999 | | 0 | 1.39 | 0 | 4.830 |
| | dual | 0.1105 | 0.1064 | 0.9001 | 0.999 | 0 | 1.5 | 2.63 | |
| 10 (CW) | single | 0.1868 | | 0.999 | | 0 | 0.9 | 0 | 10.93 |
| | dual | 0.0935 | 0.0934 | 0.9989 | 0.999 | 0 | 0.9 | 0 | |
| 13 (CCW) | single | 0.1767 | | 0.999 | | 0.07 | 1.02 | 0 | 8.710 |
| | dual | 0.1703 | 0.1702 | 0.4565 | 0.999 | 0.09 | 1.33 | 2.71 | |
| 14 (CW) | single | 0.2434 | | 0.9903 | | 0 | 0.97 | 0 | 10.77 |
| | dual | 0.2432 | 0.0002 | 0.9903 | 0.9904 | 0 | 0.97 | 0 | |
| 15 (CCW) | single | 0.2588 | | 0.999 | | 0.09 | 0.92 | 15 | 10.81 |
| | dual | 0.1295 | 0.1294 | 0.9989 | 0.999 | 0.09 | 0.92 | 15 | |
| 17 (CCW) | single | 0.265 | | 0.999 | | 0.39 | 0.56 | 0 | 9.540 |
| | dual | 0.1413 | 0.1412 | 0.9989 | 0.999 | 0 | 0.67 | 0 | |
| 29 (CCW) | single | 0.2031 | | 0.999 | | 0.11 | 0.5 | 0 | 9.380 |
| | dual | 0.1695 | 0.1694 | 0.7184 | 0.999 | 0.21 | 0.68 | 0 | |
| 30 (CCW) | single | 0.1599 | | 0.999 | | 0.09 | 0.5 | 0 | 10.87 |
| | dual | 0.0801 | 0.08 | 0.9989 | 0.999 | 0.09 | 0.5 | 0 | |
| 31 (CW) | single | 0.28 | | 0.985 | | 0 | 0.79 | 0 | 10.76 |
| | dual | 0.2798 | 0.0002 | 0.985 | 0.9851 | 0 | 0.79 | 0 | |

Coriolis force adaptation, it needs to be interpreted with caution given a number of limitations of the present study. First, the present Coriolis force adaptation paradigm could not measure active compensation without error feedback during the washout phase. Most of the studies that uncovered dual-rate processes used error clamps to exclude error feedback, but this would be a serious technical challenge for a rotating room environment. In fact, to our knowledge, none of the previous Coriolis force adaptation studies have worked with error clamps. As a surrogate solution, we deterred participants' learning from visual error feedback in the wash-out phase by blocking vision (using shutters). Yet, the experimental setup had no provision to cancel out the proprioceptive feedback of the movement, which may have driven part of the adaptation process, thereby reducing the power to observe spontaneous recovery based on

dual-rate learning and retention. Second, we quantified compensation based on reach kinematics, which are the result of an interaction between active compensation and the forces generated by the room rotation. By design, we did not have a subject specific measure of the naïve reach error during the rotation in block B. Other studies, for instance prism adaptation studies, have used movement kinematics to demonstrate spontaneous recovery, but without perturbing environmental forces [2]. As a result, the observed errors are a direct representation of the perturbation estimate. Third, due to the relatively long breaks between the different blocks of the paradigm, needed to regulate room rotation, forgetting of learned compensation may have happened [8,19]. Indeed, parameter $b$ in our model, accounting for this effect, was not always zero. In particular, the break between phase B and the washout block could have diminished the contribution of the fast process, probing only the remnants of the slow process during the washout, masking the subtle differences between the single- and dual-rate process. With these reservations in mind, let us further discuss the implications of our results.

The results of the modeling were most in line with a single-rate learning process that mediates adaptation to Coriolis forces. Although the dual(multi)-rate model is well established as an account of motor adaptation, exceptions have been reported. For instance, Ingram and colleagues (2011) showed that adaptation to familiar dynamics, for instance the dynamics of a hammer-like object, is better explained by a single- than a dual-rate model. The authors suggested that adaptation to familiar dynamics entails a change in the parameterization of a known structure [20,21], rather than the learning of the structure of the novel dynamics themselves [12]. This re-parameterization can occur much quicker than the learning of an entirely new structure. Since we encounter Coriolis forces in everyday life, for example when we reach while rotating our torso, the dynamics of these forces could likewise be very familiar to us. Additional evidence that Coriolis force adaptation might involve re-parameterization is provided by the very fast initial learning (only few trials) [13].

Furthermore, most of the Coriolis forces that we encounter are self-generated, i.e., we actively rotate our body while reaching out [22]. Therefore, we may assign perturbations due to Coriolis forces to changes of our own body rather than to changes in the world. Adapting to changes of the world might only be relevant in specific situations, e.g., when specific cues are present [23–25], while adapting to changes of our own body is always relevant and independent of context [26]. Note that in Coriolis force adaptation, the perturbations also occur independent of context, in contrast to the handle providing context in a robotic manipulandum [27,28]. This makes the assumption that the perturbation originates from the body a plausible conjecture [13]. The inferred source of a perturbation has been suggested to affect the parameterization of adaptation [29]. Perturbations with an internal cause, for example due to fatigue, could be more gradually changing and more long lasting than externally generated perturbations, for example a gust of wind. Adapting to these small internal changes allows for fast learning and minimal forgetting, which is what we see in the parameters of our single-rate model predictions.

Classically, the fast state of the dual-rate model is associated with a high learning rate and a low retention factor. However, our single-rate fits show relatively high learning rates combined with high retention factors in all participants (see Table 1). Especially the high retention rates may be responsible for the fast and full reduction of error in our subjects [30]. In line with these high retention factors, participants in the control group still retained the learned behavior after a break of more than 90 seconds. To further investigate whether Coriolis force adaptation is associated with a single adaptation process with a high retention factor, and thus results in more long lasting changes of motor control, one could increase the inter-trial-intervals and see whether trial to trial forgetting is lower for Coriolis force adaptation as compared to context dependent, and world referenced types of motor adaptation [31]. Investigating the de-

adaptation and savings in Coriolis force adaptation could further strengthen or challenge the evidence for a single-rate model in Coriolis force adaptation.

In summary, we investigated whether reach adaptation to Coriolis forces is governed by a single- or dual-rate model. Given the limitations of our study, our results suggest that a single-rate model provided a more parsimonious account of this adaptation process, perhaps because the Coriolis forces relate to familiar body dynamics and are assigned to an internal cause.

## Supporting information

**S1 Fig. Data and model fits of participants (Pattern 1).** Same legend as in Fig 4. Dual-rate model fits predicting the same pattern as the single rate model fits by setting one of the two states to zero.
(TIF)

**S2 Fig. Data and model fits of participants (Pattern 2).** Same legend as in Fig 4. Dual-rate model fits predicting the same patterns as the single rate models fit by setting both states equal.
(TIF)

**S3 Fig. Data and model fits of participants (Pattern 3).** Same legend as in Fig 4. Model fits of the dual-rate model showing typical dual-rate pattern.
(TIF)

## Acknowledgments

We would like to thank our technical service group (TSG) for the design and construction of the rotating room environment, and their excellent technical support throughout this project.

## Author Contributions

**Conceptualization:** Judith L. Rudolph, Janny C. Stapel, Luc P. J. Selen, W. Pieter Medendorp.

**Data curation:** Judith L. Rudolph.

**Formal analysis:** Judith L. Rudolph.

**Funding acquisition:** Janny C. Stapel, W. Pieter Medendorp.

**Investigation:** Judith L. Rudolph, Janny C. Stapel.

**Methodology:** Judith L. Rudolph, Janny C. Stapel, Luc P. J. Selen, W. Pieter Medendorp.

**Project administration:** Judith L. Rudolph.

**Resources:** Judith L. Rudolph, Janny C. Stapel, Luc P. J. Selen, W. Pieter Medendorp.

**Software:** Judith L. Rudolph, Luc P. J. Selen.

**Supervision:** Luc P. J. Selen, W. Pieter Medendorp.

**Validation:** Judith L. Rudolph, Luc P. J. Selen.

**Visualization:** Judith L. Rudolph.

**Writing – original draft:** Judith L. Rudolph.

**Writing – review & editing:** Janny C. Stapel, Luc P. J. Selen, W. Pieter Medendorp.

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
