## [Decision Letter · Decision Letter 0]

2 Jul 2020

PONE-D-20-15875

Single versus Dual-Rate Learning when Exposed to Coriolis Forces during Reaching Movements

PLOS ONE

Dear Dr. Rudolph,

Thank you for submitting your manuscript to PLOS ONE. After careful consideration, we feel that it has merit but does not fully meet PLOS ONE’s publication criteria as it currently stands. Therefore, we invite you to submit a revised version of the manuscript that addresses the points raised during the review process.

We look forward to receiving your revised manuscript.

Kind regards,

Chris Connaboy

Academic Editor

PLOS ONE

Additional Editor Comments:

Please pay special attention to the major issues highlighted by both reviewers and provide stronger rationale for the question(s) addressed and justifications for sample size, model fit and potential bias.

Journal Requirements:

2. Please address the following:

- Please ensure you have thoroughly detailed the recruitment procedure, including any exclusion and inclusion criteria.

- Please provide further details of participant consent, including whether or not consent was informed.

Reviewers' comments:

Reviewer's Responses to Questions

**Comments to the Author**

1. Is the manuscript technically sound, and do the data support the conclusions?

Reviewer #1: Partly

Reviewer #2: Yes

2. Has the statistical analysis been performed appropriately and rigorously? 

Reviewer #1: No

Reviewer #2: Yes

3. Have the authors made all data underlying the findings in their manuscript fully available?

Reviewer #1: Yes

Reviewer #2: Yes

4. Is the manuscript presented in an intelligible fashion and written in standard English?

Reviewer #1: Yes

Reviewer #2: Yes

5. Review Comments to the Author

Reviewer #1: The authors investigated whether adaptation to Coriolis forces is governed by a single or dual-rate adaptation process. To do so, they let participants perform a pointing task in a room that could rotate either clockwise or counterclockwise, introducing Coriolis forces on the arm. The manuscript reads as very careful work in which much attention has been paid to the design of the experiment and to parts of the statistical analysis. Yet, for me as someone who studied the dual-rate process, it leaves two pressing questions unanswered. First: did the authors have enough power to make a between-groups comparison of dual-rate model fits with two additional parameters? Second, and related: recent work by Albert & Shadmehr 2018 (Journal of Neurophysiology) has shown statistical problems with fitting the dual rate model. To what extent did the procedure used by the authors fix these problems? As I believe that these issues need to be addressed before being able to interpret the data as evidence that adaptation to Coriolis forces is governed by a single-rate process, I have suggested a major revision.

Major comments

Sample size

In my experience, adaptation is highly variable between participants. Often, I enthusiastically presented the data from 9 participants at a conference, having to change my conclusions when more participants were tested. The sample size of 10 participants in the control group and 17 participants in the experimental group seems quite small to me. The standard errors in Figure 2 suggest that participants within a group behaved relatively consistently, which is re-assuring. However, I’d like to see more evidence that the data were normally distributed around the mean. This is important for the interpretation of the adaptation curves presented in figure 3.

Robustness of model fits

I am also concerned about the robustness of the model fits. I couldn’t find information on the quality of the fits: confidence intervals on the parameters in table 1, for instance. I never worked with the Bayesian Information Criterion, but if both models fitted the data poorly due to a high level of noise, wouldn’t the model with fewer parameters win? In that case, the finding that the BIC was higher for the single-rate model could be an artifact of the data being noisy. Without more evidence on the statistical reliability of the model fits, adaptation to Coriolis forces could also be governed by a dual rate process with a very slow ‘slow process’, I think. Of course, a fundamental question is when the slow process is so slow it can be disregarded.

Potential bias

The text mentions that the data converge to zero in Figure 3. Looking at Figure 3, the data do not seem to converge towards zero. For CW-CCW they do, but not for CCW-CW. If there is a bias in the data with either leftward or rightward errors being more likely, this could have affected the results in the washout phase.

Minor

Line 25: I would phrase the sentence a little bit differently. Many types of errors contribute to motor adaptation (e.g. target errors, performance errors, sensory prediction errors). All these errors reflect ‘mistakes’.

line 86; please mention how many participants were excluded due to failure to follow task instructions and how they failed to follow the task instructions

Line 131: why were these movement times chosen?

Line 162: how was it determined whether more practice trials were needed?

Line 201: Why was a cartesian coordinate system used? Compensating for a rotation would suggest using a spherical coordinate system to me. That way the lateral error could be expressed in degrees azimuth

205 Does the y-position refer to the sagittal position? The y-postition can refer to multiple axes

The LD abbreviation is unnecessary

Line 241: please mention that the fit procedure is explained later

336: It isn’t clear whether the washout block or the model fits are the crucial part of the data analysis

Figure 2. Having the figures and text separately, Figure 2 was difficult to grasp with all the different colours.

Line 402: spontaneous recovery of the adaptation to perturbation A?

Figure 3: The data in block B are very difficult to see. Where is the SE?

Line 430: This sentence is difficult to understand. What do the authors refer to?

Line 439: Please specify what the typical pattern would be

Line 443: Please present the data such that this can be seen in the figure

Figure 4: It would help me to provide column titles in the figure

Reviewer #2: I commend the authors on an original and well-designed study that is novel and relevant to the field. The manuscript presents a study in which participants perform a reaching task while being exposed to Coriolis forces. The adaptation process is described in the different phases of the experiment and two models are proposed to explain the learning process: a single rate and a dual rate model, of which the single rate model seems to have the best fit. The authors do well in highlighting some limitations to the study and to discuss the meaning of these results in light of previous studies. Overall, the manuscript is of good quality, but some components of the methods and results section lack clarity in their reporting.

Major issues

1. I feel some information is presented in an order that breaks up the flow of the manuscript. A statement of the main research question seems to be subtly mentioned in line 61-65 without stating any hypotheses or operationalization. After this the introduction is finished with a summary of the methods and the results section. An operationalization of the research question does seem to be mentioned in the results, with the paragraphs on line 386-408. I believe it would improve the readability of the manuscript if this part of the results would be moved to the introduction and that the introduction would end with a clear statement of the aims or research question.

Minor issues

1. Line 108 (Fig 1.). The grey area in this figure is hard to see (both printed and on screen). Please use a darker shade of grey.

2. Line 121-122. On the next page it is introduced that this statement is only true for about half the trials. Please report both procedures as to the opening and closing of the glasses here.

3. Line 125. The figure caption for Fig 1C mentions that vision is occluded during the reaches, but this sentence indicates participants need to locate the target prior to the forward movement when shutters are closed. Please clarify when shutters were closed.

4. Line 131-136 describe the required timing of the reaching movement. What happened with trials that were not inside the time limit? Were these removed? Was there any procedure in place in case participants were fast or slow repeatedly?

5. Line 138-139 mentions a change in protocol halfway through the study (i.e. timing of the occlusion goggles), which due to the non-random group allocations almost overlaps with the separation between experimental and control group. Can the authors discuss whether they think this limitation might have influenced the outcomes?

6. Line 139 mentions the percentage of trials with occlusion errors. What happened to these trials? Were they deleted and if not, did they lead to outliers in the data?

7. Line 193-195. What was done with the data from the three control participants with vision of the start of their backward movement. If this data was used, were these outliers?

8. Line 233-235. Do ‘A’ and ‘B’ here refer to ‘Block A’ and ‘Block B’? Please specify.

6. PLOS authors have the option to publish the peer review history of their article (what does this mean?). If published, this will include your full peer review and any attached files.

Reviewer #1: No

Reviewer #2: **Yes: **Steven van Andel

---

## [Author Response · Author response to Decision Letter 0]

28 Jul 2020

For responses to specific reviewer and editor comments please see uploaded documents (CoverLetter_PlosOne_Revisions and Response to Reviewers).

---

## [Decision Letter · Decision Letter 1]

2 Sep 2020

PONE-D-20-15875R1

Single versus Dual-Rate Learning when Exposed to Coriolis Forces during Reaching Movements

PLOS ONE

Dear Dr. Medendorp,

Thank you for submitting your manuscript to PLOS ONE. After careful consideration, we feel that it has merit but does not fully meet PLOS ONE’s publication criteria as it currently stands. Therefore, we invite you to submit a revised version of the manuscript that addresses the points raised during the review process.

We look forward to receiving your revised manuscript.

Kind regards,

Chris Connaboy

Academic Editor

PLOS ONE

Additional Editor Comments (if provided):

The reviewers both highlight that the work has improved substantially, but each require some further very minor changes to clarify and improve the work further.

Reviewers' comments:

Reviewer's Responses to Questions

**Comments to the Author**

1. If the authors have adequately addressed your comments raised in a previous round of review and you feel that this manuscript is now acceptable for publication, you may indicate that here to bypass the “Comments to the Author” section, enter your conflict of interest statement in the “Confidential to Editor” section, and submit your "Accept" recommendation.

Reviewer #1: All comments have been addressed

Reviewer #2: (No Response)

2. Is the manuscript technically sound, and do the data support the conclusions?

Reviewer #1: Yes

Reviewer #2: Yes

3. Has the statistical analysis been performed appropriately and rigorously? 

Reviewer #1: Yes

Reviewer #2: Yes

4. Have the authors made all data underlying the findings in their manuscript fully available?

Reviewer #1: Yes

Reviewer #2: Yes

5. Is the manuscript presented in an intelligible fashion and written in standard English?

Reviewer #1: Yes

Reviewer #2: Yes

6. Review Comments to the Author

Reviewer #1: The authors have adequately responded to my questions. I do have some additional minor comments most of which relate to adding nuance to the conclusion that adaptation is governed by a single-rate process. I think one could alternatively conclude that adaptation is governed by a fast process. In other words: a difference in the system used or in the parameters used.

1a. What would be the minimal contribution of a slow process that could be detected with the model comparison?

I think it would be interesting to simulate predictions of the two models with expected parameters (taking slow and fast state values from the literature) and with the observed level of noise. Would the dual rate model win the BIC comparison based on the simulated data?

1.b Interpretation of BIC comparison

There might be some nuance to the interpretation of the BIC comparison. When the model with fewer parameters wins, does this mean that this model provides a significantly better account of the data (line 532) or does it mean that adding two parameters did not result in better information?

1.c Interpretation: single rate versus efficient fast process

In the discussion the authors might provide nuance to adaptation either being governed by a dual- or single-rate process. One could alternatively argue that adaptation is always governed by a multi-rate process in which the contribution of the slow rate depends on how fast the fast state can learn. When the fast state can learn very quickly, there are little errors left to learn from for the slow state.

In this respect, it is interesting that the learning rates for the fast process do not seem very high, for instance compared to my own work (van der Kooij et al., 2015 PLOS). The fact that errors are reduced quickly seems to be due to the high retention rates rather than to the high learning rates. Consistently, the authors find that asymptotic adaptation is complete whereas many studies in the literature report incomplete asymptotic adaptation, which can be explained by incomplete retention.

2. Prediction on the control group versus experimental group.

I couldn’t find a clear prediction on the expected difference between the control group and experimental group before the presentation of the results. For instance on line 348 it would be helpful to add a prediction for the washout phase.

3.

Line 320: The authors report that 200 model fits were made using randomly selected starting values. How were these 200 fits converged into a single set of fit parameters?

Reviewer #2: I commend the authors again on a well written study, with my apologies for taking long to complete this review. The revised manuscript reads well and the revisions address most of my concerns. In fact, I have only two minor issues left, which relate to my earlier point that did not come across clearly in the first review. These relate to the protocol for closing the PLATO spectacles

Line 123-126. This sentence does not accurately summarize the spectacle-closing protocol, as it implies that all spectacle-closing was based on contact with the touch screen, but on line 147-150 we learn this is not the case. I would suggest making some mention of these separate protocols here, and/or, to move up the paragraph on line 143-150 to this point so all spectacle-related information is presented together. Furthermore, later in the text, in line 196-204, it is introduced that a different protocol is used in the washout block, so please mention for which blocks this particular protocol is used.

Line 125. Fig 1A does not seem to mention the PLATO spectacles, should this be Fig 1C?

7. PLOS authors have the option to publish the peer review history of their article (what does this mean?). If published, this will include your full peer review and any attached files.

Reviewer #1: No

Reviewer #2: **Yes: **Steven van Andel

---

## [Author Response · Author response to Decision Letter 1]

7 Sep 2020

Revision PONE-D-20-15875 R1 “Single versus Dual-Rate Learning when Exposed to Coriolis Forces during Reaching Movements” by Rudolph et al. 

We thank the reviewers for their re-evaluations. Below, you will find the reviewers’ remaining comments, and our reply in bold. We hope that our manuscript is now acceptable for publication. 

Reviewer #1: The authors have adequately responded to my questions. I do have some additional minor comments most of which relate to adding nuance to the conclusion that adaptation is governed by a single-rate process. I think one could alternatively conclude that adaptation is governed by a fast process. In other words: a difference in the system used or in the parameters used.

Response: We are not sure we completely understand the comment. If we conclude that Coriolis force adaptation is governed by a fast process, that is also a single-rate process. 

1a. What would be the minimal contribution of a slow process that could be detected with the model comparison?

I think it would be interesting to simulate predictions of the two models with expected parameters (taking slow and fast state values from the literature) and with the observed level of noise. Would the dual rate model win the BIC comparison based on the simulated data?

Response: This is an interesting point but a study on its own; see for example Albert and Shadmehr (2018) to appreciate the complexities involved and assumptions needed to perform such an analysis. For the present study, it requires us to make too many different assumptions, including an estimation of the individual noise levels in the single-rate and dual rate process, the learning and retention rates of the processes, which we feel does not really make an important contribution to our work. Our goal has been to present the data, obtained using a carefully designed paradigm, and perform model fitting to the data using well-established methods. We have been extremely careful, as conveyed by many explicit sentences, to not over-interpret our conclusions - adding further modeling simulations would not allow us to make stronger claims. Note, for the reviewer’s interest, we did run BIC comparisons between single-rate and dual-rate fits on an existing spontaneous recovery dataset from a force field adaptation paradigm, showing evidence for the latter. 

1.b Interpretation of BIC comparison

Response: There might be some nuance to the interpretation of the BIC comparison. When the model with fewer parameters wins, does this mean that this model provides a significantly better account of the data (line 532) or does it mean that adding two parameters did not result in better information?

We understand this point; we regard the BIC analysis as a maximum likelihood estimate driven method of assessing the model fits penalized for the number of free parameters. Using this analysis, gave us strong to very strong evidence in favor of the single-rate model fits. However, the BIC does not make any statement about the goodness of fit and in theory it could be that both the single- and dual rate model do not fit the data well. That’s the reason that we also present R2-values. These values indicate reasonable fits, that do not really differ between the models, as we stated in the revised manuscript. To meet the reviewer’s request of adding more nuance we now changed ‘better account’ into ‘a more parsimonious description” at several places in the MS. Furthermore, we realize that the use of ‘significantly’ in line 532 was ill chosen and have also replaced this by “more parsimonious”.

1.c Interpretation: single rate versus efficient fast process

In the discussion the authors might provide nuance to adaptation either being governed by a dual- or single-rate process. One could alternatively argue that adaptation is always governed by a multi-rate process in which the contribution of the slow rate depends on how fast the fast state can learn. When the fast state can learn very quickly, there are little errors left to learn from for the slow state.

In this respect, it is interesting that the learning rates for the fast process do not seem very high, for instance compared to my own work (van der Kooij et al., 2015 PLOS). The fact that errors are reduced quickly seems to be due to the high retention rates rather than to the high learning rates. Consistently, the authors find that asymptotic adaptation is complete whereas many studies in the literature report incomplete asymptotic adaptation, which can be explained by incomplete retention.

Response: We agree that only if the fastest learning process has a high enough retention rate it can fully compensate for the perturbations and reduce error on its own. As a result, slower processes won’t be able to contribute anymore. We do not want to claim that there are no slower processes, but at least they are not prominent enough to show up in our data and analysis. Even proponents of the popular dual-rate model won’t say that there are no slower processes at play, but that in the specific tasks that they investigate they don’t have a significant influence. We feel that we have already provided sufficient nuance, and added our reservations at several places, in particular the discussion. We now make a reference to Van der Kooij et al.’s work when we discuss the learning and retention rates of our single-rate fits (see l. 601-604 )

2. Prediction on the control group versus experimental group.

I couldn’t find a clear prediction on the expected difference between the control group and experimental group before the presentation of the results. For instance on line 348 it would be helpful to add a prediction for the washout phase.

Response: Done. We added that we expect no rebound effects in their washout phase (see l. 353-354 )

3. Line 320: The authors report that 200 model fits were made using randomly selected starting values. How were these 200 fits converged into a single set of fit parameters?

Response: We added this information (see l. 324). Across the 200 fits, we selected the set of parameters that yielded the smallest mean squared error. 

Reviewer #2: I commend the authors again on a well written study, with my apologies for taking long to complete this review. The revised manuscript reads well and the revisions address most of my concerns. In fact, I have only two minor issues left, which relate to my earlier point that did not come across clearly in the first review. These relate to the protocol for closing the PLATO spectacles

Line 123-126. This sentence does not accurately summarize the spectacle-closing protocol, as it implies that all spectacle-closing was based on contact with the touch screen, but on line 147-150 we learn this is not the case. I would suggest making some mention of these separate protocols here, and/or, to move up the paragraph on line 143-150 to this point so all spectacle-related information is presented together. Furthermore, later in the text, in line 196-204, it is introduced that a different protocol is used in the washout block, so please mention for which blocks this particular protocol is used.

Response: We have clarified the initially text. We prefer to bring up the whole paragraph on line 143, given that it also contains information unrelated to the spectacles, which needs to be presented first. We hope that the current adjustment made the spectacle-closing protocol sufficiently clear now (see l.125-128). 

Line 125. Fig 1A does not seem to mention the PLATO spectacles, should this be Fig 1C?

Response: The reviewer is right. We have corrected the referencing.

---

## [Editor Report · Decision Letter 2]

1 Oct 2020

Single versus Dual-Rate Learning when Exposed to Coriolis Forces during Reaching Movements

PONE-D-20-15875R2

Dear Dr. Medendorp,

We’re pleased to inform you that your manuscript has been judged scientifically suitable for publication and will be formally accepted for publication once it meets all outstanding technical requirements.

Kind regards,

Chris Connaboy

Academic Editor

PLOS ONE
---

## [Editor Report · Acceptance letter]

8 Oct 2020

PONE-D-20-15875R2 

Single versus Dual-Rate Learning when Exposed to Coriolis Forces during Reaching Movements 

Dear Dr. Medendorp:

I'm pleased to inform you that your manuscript has been deemed suitable for publication in PLOS ONE. Congratulations! Your manuscript is now with our production department. 

Kind regards, 

on behalf of

Dr. Chris Connaboy 

Academic Editor

PLOS ONE